# Treatment with Beta-Blockers and ACE-Inhibitors in Breast Cancer Patients Receiving Adjuvant Trastuzumab-Based Therapy and Developing Mild Cardiac Toxicity: A Prospective Study

**DOI:** 10.3390/cancers12020327

**Published:** 2020-01-31

**Authors:** Elena Geuna, Pasquale Lombardi, Rossella Martinello, Davide Garino, Alessandro Bonzano, Danilo Galizia, Annamaria Nuzzo, Paola Berchialla, Paolo Becco, Monica Mangioni, Lorena De Zarlo, Filippo Montemurro

**Affiliations:** 1Multidisciplinary Oncology Outpatient Clinic, Candiolo Cancer Institute, FPO-IRCCS, 10060 Candiolo, Italy; elena.geuna@ircc.it (E.G.); danilo.galizia@ircc.it (D.G.); 2Medical School, University of Turin, 10124 Turin, Italy; pasquale.lombardi@ircc.it (P.L.); davide.garino693@edu.unito.it (D.G.); paolo.becco@ircc.it (P.B.); 3Oncologia Medica, Ospedale Cardinal Massaia, 14100 Asti, Italy; rmartinello@asl.at.it; 4Cardiology Unit, Candiolo Cancer Institute, FPO-IRCCS, 10060 Candiolo, Italy; alessandro.bonzano@ircc.it; 5Clinical Research Office, Candiolo Cancer Institute, FPO-IRCCS, 10060 Candiolo, Italy; annamaria.nuzzo@ircc.it; 6Dipartimento di Scienze Cliniche e Biologiche, University of Turin, 10124 Turin, Italy; paola.berchialla@unito.it; 7Clinical Laboratory, Candiolo Cancer Institute, FPO-IRCCS, 10060 Candiolo, Italy; monica.mangioni@ircc.it (M.M.); lorena.dezarlo@ircc.it (L.D.Z.)

**Keywords:** breast cancer, trastuzumab, cardiac toxicity, enalapril, carvedilol, ACE inhibitors, beta blockers

## Abstract

Background: Angiotensin Converting Enzyme inhibitors (ACEis) and beta-blockers (BB) are suggested to prevent and treat trastuzumab-related cardiac toxicity. We performed a prospective clinical trial in women experiencing mild cardiac toxicity (MCT) while on adjuvant treatment with trastuzumab. Methods: MCT was defined as an asymptomatic absolute decrease in LVEF of ≥ 10 percentage units to >50%. Treatment consisted of enalapril 2.5 mg bid and carvedilol 3.75 mg bid, which were up-titrated to 10 mg bid for the enalapril and 6.25 mg bid of carvedilol. In patients receiving study drug, the primary study end-point was LVEF recovery, which was defined as a post-trastuzumab LVEF returning to no less than −5 percentage points of the baseline value. Results: 103 patients were enrolled, 100 started trastuzumab, and 98 completed the planned treatment. Sixteen patients (16%) had MCT and received study drugs until trastuzumab completion. None of these patients achieved a post-trastuzumab LVEF recovery. Nevertheless, treated patients had significantly higher median LVEF recovery from nadir to post-trastuzumab LVEF in (8% points vs. 4% points, respectively, *p* = 0.004), resulting in no difference in post-treatment LVEF values compared to patients without MCT. Conclusion: Treatment of MCT with ACEis and BB allows faster LVEF recovery from nadir values and should be further studied in this setting.

## 1. Introduction

Amplification or overexpression of the human epidermal growth factor receptor 2 (HER2) oncogene characterizes approximately 15–20% of patients with breast cancers (BC) and is associated with early disease progression and poor prognosis [1,2].

Trastuzumab, a recombinant humanized monoclonal antibody that binds the extracellular domain of HER2, improves overall survival (OS) and disease-free survival (DFS) in patients with early, HER2 positive operable breast cancer [3].

Although trastuzumab is generally well tolerated, cardiac dysfunction is a significant adverse effect, especially in patients with prior or concomitant exposure to anthracycline-based chemotherapy [4]. In a recent pooled analysis of three pivotal randomized clinical trials, 11.3% of the women in the trastuzumab arms experienced a cardiac event, which was mildly symptomatic or asymptomatic for the majority of them (8.7%) [5]. Severe congestive heart failure occurred in 2.8% of the patients receiving trastuzumab.

Trastuzumab-related cardiac toxicity may lead to definitive interruption of trastuzumab and long-term cardiac sequelae. For this reason, especially in the early breast cancer setting, guidelines on how to treat cardiac dysfunction during treatment have been widely adopted [4]. Practically, while symptomatic cardiac toxicity mandates trastuzumab interruption and appropriate cardiac therapy, thresholds have been identified in asymptomatic patients experiencing left-ventricular ejection fraction losses compared to pre-treatment values to trigger temporary interruption and cardiac treatments.

Clinical studies have addressed strategies to predict, prevent, and treat clinically significant, trastuzumab-related cardiac toxicity [6]. Cardiac markers, such as high sensitive troponin I (hs-TnI) and N-Terminal Prohormone of Brain Natriuretic Peptide (NT-ProBNP), are sensitive and specific markers to detect myocardial injury, to predict treatment-related decreases in left ventricular ejection fraction (LVEF) and the development of left ventricular dysfunction and heart failure [7]. Based on the physiopathology of trastuzumab-related cardiac toxicity, angiotensin-converting enzyme inhibitors (ACEis) and beta-blockers (BB) are widely recommended as initial treatment of heart failure and for clinically significant trastuzumab–related cardiac dysfunction because of their demonstrated ability to attenuate or reverse left ventricular remodelling [6,7,8].

Loss of LVEF percentage points not requiring specific intervention (i.e., not fulfilling the criteria for trastuzumab interruption), occurs but is largely underreported in the clinical practice. Currently, it is not fully understood whether this “mild” cardiac toxicity (MCT) has immediate or late consequences. We, therefore, decided to conduct a prospective trial evaluating the potential role of early pharmacological intervention with ACEis and BB in patients with operable, HER2 positive breast cancer who, during trastuzumab treatment, developed LVEF loss not requiring trastuzumab discontinuation.

## 2. Results

### 2.1. Patient Disposition

A total of 103 women with HER2-positive localized breast cancer signed informed consent and entered the study. Table 1 shows the demographics of the enrolled patients, and Appendix A shows the protocol flow.

One patient who had signed the informed consent was found to have stage IV during the subsequent diagnostic workup and was excluded from subsequent analyses. Of 102 patients starting adjuvant treatment, one developed metastatic disease upon the completion of anthracycline-based chemotherapy. We recorded the pre-trastuzumab LVEF value, but this patient, who started chemotherapy plus trastuzumab and pertuzumab as routine first-line treatment for advanced breast cancer, was excluded from subsequent analyses. Another patient, upon the completion of anthracycline-based therapy, decided to continue treatment at another Institution and dropped off protocol. Of a total of 100 patients eligible for trastuzumab, 98 had a pre-trastuzumab LVEF assessment performed in the protocol-defined time-window, whereas two who had received anthracycline-based therapy, had a pre-trastuzumab LVEF performed after the initiation of trastuzumab-based therapy. These patients will be included in the intent-to-treat analyses. Eighty-one patients started trastuzumab with taxanes after 3 or 4 cycles of anthracycline (either FEC_100_ X3 or EC_90_ X4) and are defined as “group 1”. The other 19 patients did not receive anthracycline before trastuzumab and are defined as “group 2”. Ninety-eight of the patients (98%) were able to complete the planned intended treatment with trastuzumab (median duration 52 weeks, range 14–76). For one patient, trastuzumab was stopped after 14 weeks because of cardiac side effects. Another patient stopped trastuzumab after 18 weeks because of personal choices, in the absence of any toxicity.

### 2.2. Cardiac Monitoring by Ultrasonography

A total of 719 LVEF evaluations were performed during the study (Appendix A. Appendix A and Figure 1 provide summary statistics of the overall LVEF findings of our study at the baseline, nadir (lowest LVEF registered during the study) and after treatment (from 3 to 12 months after the last dose of trastuzumab).

Notably, before trastuzumab, group 1 patients had significantly lower values than group 2 patients (*p* = 0.048). Nadir values were significantly lower to both baseline and pre-trastuzumab values overall and separately for group 1 and 2 patients. While not different between group 1 and 2 patients, post-trastuzumab values were significantly lower compared to baseline values.

### 2.3. Cardiac Events

Nineteen patients were referred to our cardiologist during the study. For 3 of them, the reasons were not related to LVEF findings (palpitations, shortness of breath, and EKG abnormalities), and trastuzumab was continued and completed as planned. The other 16 patients (15 in group 1 and 1 in group 2) developed mild cardiac toxicity (MCT) and were started on study drugs (16%, 95% C.I. 9–25%). These were well tolerated during the up-titration phase, with no patien discontinuing treatment because of intolerance. One of these patients developed further LVEF loss to less than 40%, became symptomatic and stopped trastuzumab treatment. The remaining 15 patients were able to complete trastuzumab without treatment withholdings or delay. Except for the patient developing symptomatic LVEF loss, for no patient LVEF value dropped below 50% during treatment (Appendix A. Figure 2 shows the Kaplan Meyer curve of time to the first presentation in the 16 patients who experienced MCT.

The median time-to MCT was 26 weeks from the initiation of adjuvant therapy (95% C.I. 13–37 weeks). Appendix A and Figure 3 provide summary statistics of the LVEF findings in patients developing and not developing MCT.

Baseline and pre-trastuzumab LVEF values between patients with or without MCT were not statistically significantly different. As expected, the difference in nadir LVEF between patients with and without the MCT was highly statistically significant. Interestingly, post-trastuzumab LVEF values were not statistically significantly different between patients with and without MCT. Indeed, patients who experienced MCT and, therefore, received study drugs had a significantly higher median LVEF recovery from nadir to post-trastuzumab LVEF than patients who did not develop MCT (8% points vs. 4% points, respectively, *p* = 0.004).

No patient developing MCT recovered, after trastuzumab completion, to an LVEF no less than −5 percentage points of the baseline value (primary study endpoint). Conversely, a recovery from nadir occurred in 48% of patients with did not develop MCT (*p* < 0.001). As a result, post-trastuzumab LVEF values were lower than baseline and pre-trastuzumab values.

### 2.4. Potential Predictors of MCT

Table 2 shows the univariate analysis of potential predictors of MCT. Age, baseline LVEF, Body Mass Index (BMI), baseline TN I and BNP values were studied both as continuous and dichotomous variables.

Continuous variables were dichotomized around their median values. Although there were trends for smoking, hypertension, and receipt of anthracycline, only a family history of cardiovascular disease (CVD) was significantly associated with a three-fold increase in the risk of MCT. Various multivariable models were studied using all the potential predictors, regardless of their significance at univariate analysis, yet a family history of CVD remained the only significant predictor.

## 3. Discussion

In our study, we found a 1% incidence of symptomatic cardiac toxicity leading to treatment discontinuation and a 98% trastuzumab completion rate with no treatment delays due to cardiac events. LVEF values declined substantially and, despite recovery in most patients, post-treatment values were significantly lower compared to baseline. No patient experiencing MCT achieved an LVEF recovery to no less than –5 percentage point of the baseline value, which was the primary study endpoint. Nevertheless, the magnitude of recovery from nadir values was significantly higher in patients receiving study drugs. Therefore, despite lower LVEF nadir values, post-trastuzumab LVEF values did not differ significantly between patients experiencing and not experiencing MCT, suggesting a potential role for early medical intervention targeted on asymptomatic LVEF loss. Univariate analysis showed that only a family history of cardiovascular disease was associated with an increased risk of MCT, although strong trends not reaching statistical significance were seen for having received anthracycline before trastuzumab and for a positive personal history of smoking. These results provide a rationale for de-escalating strategies and a randomized study of early targeted pharmacological intervention in women receiving trastuzumab-based adjuvant therapy for breast cancer.

Based on the OHERA trial, the largest, prospective, observational study of cardiac toxicity in patients receiving adjuvant trastuzumab, the incidence of cardiac events in a “real world” population is in the range of that reported in the pivotal clinical trials [9].

Beyond a 2.6% incidence of clinically evident congestive heart failure (Class II–IV according to the New York Heart Association), about 8% of the patients experienced an asymptomatic, but significant LVEF drop of >10% to below 50%. Interestingly, only 67% of these patients, at a median follow-up of 5 years, had an LVEF recovery. Finally, the incidence of mild cardiac toxicity not leading to discontinuation of trastuzumab was not reported (i.e., >10% but not to below 50%). In general, this type of LVEF drop is not reported in clinical trials and real-world studies.

A large analysis with 13 years of follow-up recently addressed this often overlooked issue. The authors focused on the implications of an early, asymptomatic LVEF drop on the subsequent development of significant or symptomatic cardiac toxicity [10]. In 337 out of 787 patients, a 5% or more LVEF loss from baseline occurring within the initial 3 months of trastuzumab treatment was predictive of the development of either a significant, asymptomatic reduction in LVEF or of congestive heart failure, which occurred in 5% and 2.3% of patients, respectively.

While this study points at early and small LVEF decreases as a predictor of “on-treatment” cardiac dysfunction, the long-term consequences of early and asymptomatic LVEF loss that, by guidelines, does not require trastuzumab interruption, are mostly unknown.

Our finding that LVEF after trastuzumab treatment is significantly lower than at the baseline is consistent with the long-term data of the PHARE trial, investigating one year vs. 6 months of adjuvant trastuzumab treatment [11]. At a 30-months follow-up cutoff, LVEF mean values, that had declined by 3.6% at the end of trastuzumab treatment, compared with baseline values, tended to recover, but for almost a half of the patients, LVEF recovery was only partial. Still, the question remains “would failure to recover to a pre-treatment LVEF value predispose patients to late cardiac events?” awaits a response. Yet, if early LVEF drop predicts on-treatment cardiac events, in a woman who has completed adjuvant trastuzumab, ageing and related conditions, use of adjuvant endocrine therapy and menopausal changes may concur, together with a compromised myocardial function, to late cardiac effects.

Due to the importance of cardiac-related events associated with trastuzumab-based adjuvant therapy, clinical trials have been conducted to assess the worth of chemoprevention with cardiac drugs in patients exposed to either anthracycline, trastuzumab or both in the adjuvant setting [12,13,14,15,16,17]. Of these, three focused specifically on trastuzumab-related cardiac toxicity, which is pathogenetically different from anthracycline-related cardiomyopathy and have been fully published [12,16,17]. Boekhout and colleagues enrolled 210 women undergoing adjuvant trastuzumab-based therapy, who were randomized to either placebo or the angiotensin-II receptor inhibitor candesartan. Study treatment was started concomitantly with trastuzumab and continued until 26 weeks from trastuzumab completion [12]. In this study, candesartan failed to show any effect on LVEF decrease or the development of cardiac toxicity and found no correlation with cardiac biomarkers (NT-proBNP and hs-TnT). The Manticore study was a three-arm, randomized study evaluating whether, compared with placebo, perindopril, or bisoprolol given for the entire duration of trastuzumab therapy could prevent trastuzumab-mediated cardiac remodelling, compared with placebo [16]. Although an effect on avoiding LVEF losses was observed, neither bisoprolol nor perindopril could prevent cardiac remodeling. In a larger, randomized, double-blind, phase III study, Gauglin and colleagues evaluated whether lisinopril or carvedilol could reduce cardiac events, compared with placebo [17]. While in the overall population the rate of cardiac toxicity and the cardiotoxicity-free interval were comparable in the three arms, patients who had been previously exposed to anthracycline and treated with either lisinopril or carvedilol experienced longer cardiotoxicity-free interval. Taken together, these studies do not strongly support the systematic use of cardiac medication in all candidates to trastuzumab-based adjuvant therapy.

Differently from the studies mentioned above, we sought to target early and mild asymptomatic LVEF reductions, which occurred in 16% of the patients. We believe that our findings concur to confirm a role for cardiac drugs to be used proactively in patients whose LVEF values show a mild decline after the start of trastuzumab-based therapy. We must, however, acknowledge some limitations of our study, which was a relatively small, non-randomized trial enrolling patients who were not already on treatment with the classes of cardiac drugs that we used. Consequently, this may have influenced the low rate of moderate or severe cardiac dysfunction that we observed, with only one patient discontinuing treatment and no patient developing an LVEF drop >10% to below 50%. Yet, a relative contribution of early intervention with cardiac drugs to this low rate clinically significant events can be assumed, also considering the results of the other available clinical trials. Furthermore, the lack of a randomized design does not allow us to conclude with confidence that the extent of LVEF recovery after nadir values was due to cardiac drugs. Indeed, an improvement from nadir values was also seen in patients who did not develop the cardiac event of interest in the absence of drugs.

On the other hand, our study also has some strengths; for example, all LVEF evaluations were performed by the same cardiologists (A.B.), using the same instrument and the same protocol. Indeed, having chosen to focus on mild LVEF changes, interobserver variability could be an issue in multicentric studies [18]. Furthermore, all patients were managed at our breast clinic, ensuring consistency of treatment procedures.

## 4. Materials and Methods

### 4.1. Study Population

CARDIORETE (EudraCT number 2011-002207-15) is a nonrandomized, open-label, interventional, single group cardiac safety study conducted in full accordance with Good Clinical Practice guidelines/Declaration of Helsinki.

All consecutive women with non-metastatic breast cancer eligible for trastuzumab-based adjuvant therapy at our Institution from 1 May 2012 to 31 September 2016 were screened for this protocol.

The main inclusion criteria were age ≥18 years, newly diagnosed, HER2-overexpressing, non-metastatic breast cancer, planned adjuvant treatment with trastuzumab, renal, hepatic and bone functions permissive for planned treatment, and ability to provide informed consent.

Exclusion criteria included baseline LVEF < 55%, active cardiological comorbidity, contraindication to, or current treatment with, ACEi or BB and uncontrolled, non-cardiac concomitant diseases. Patients were planned to receive trastuzumab intravenously, according to international protocols, for one year. No patient received pertuzumab added to trastuzumab, because this drug is not funded in Italy for patients with early, HER2-positive breast cancer.

The reference Ethics Committee approved the protocol (Comitato Etico Interaziendale Ospedale San Luigi Gonzaga, Regione Gonzole 10, 10043, Orbassano, Italy, determination number 147/2011 of 17/Oct/2011) and written informed consent was obtained from all patients before the initiation of neo- or adjuvant therapy.

### 4.2. Study Objectives

The primary objective of this study was to evaluate the effect of early intervention with ACE inhibitors and beta-blockers in women developing mild cardiac toxicity (MCT), defined as an asymptomatic absolute decrease in LVEF of ≥ 10 units from baseline to no less than 50%. In these circumstances, widely adopted algorithms suggest trastuzumab continuation and no indication of pharmacological treatments [4].

Secondary objectives included the study of risk factors for MCT in a population of patients undergoing adjuvant, trastuzumab-based therapy, including baseline levels of plasma BNP (b-type natriuretic peptide) and TnI. Included in our original objectives, but not reported in this paper were; (1) the evaluation of fluctuations of BNP and TnI during treatment, and (2) the study of baseline ultrasonographic parameters that could predict MCT. These two latter objectives will be reported in a further manuscript.

### 4.3. Study Procedures

LVEF measurement by echocardiography was performed at baseline (before chemotherapy and/or trastuzumab administration), every 3 months during trastuzumab therapy and at 6 and 12 months after the completion of the scheduled treatment. Patients with MCT were identified by the cardiologist during LVEF examination, placed on study drugs and followed up every two weeks during drugs up-titration. Once reached target drug doses, patients were referred to the the cardiologist outside the scheduled LVEF assessments only if clinically indicated. In patients lost to follow-up or who died for breast cancer, the last evaluation was considered as the final measurement.

In patients developing MCT, treatment with enalapril 2.5 mg bid and carvedilol 3.75 mg bid was initiated and up-titrated to the maximum established dose (10 mg bid for the enalapril and 6.25 mg bid of carvedilol). Study drugs were administered during adjuvant trastuzumab treatment and continued for three weeks after the last trastuzumab infusion, provided that patients were still asymptomatic and with LVEF within normal Institutional limits (≥55%). Otherwise, cardiac drugs could be continued if clinically indicated. Since there is no recommendation to treat with ACE inhibitors and beta-blockers patients with mild cardiac toxicity as it was defined in our paper, we decided to limit the administration of study drugs to the completion of adjuvant trastuzumab. Patients with significant cardiotoxicity were treated according to guidelines, including trastuzumab temporary or definitive discontinuation and pharmacological interventions.

### 4.4. Cardiac Biomarker Evaluation

Plasma BNP (b-type natriuretic peptide) and Troponin I (TnI) concentration were determined before chemotherapy treatment start, before trastuzumab start, and at each cardiologic check during and after the completion of the therapy. Levels of BNP were determined by the Triage BNP test (Beckman Coulter, Inc. Brea, CA, USA) using the Beckman Access DxI 800 platform. The upper limit of normal was 100 pg/mL. Levels of TnI were determined by the Access AccuTnI immunometric assay (Beckman Coulter) using the Beckman DxI 800 platform. The upper limit of normal was 0.05 ng/mL. Determinations were carried on in our laboratory according to the manufacturer’s instructions.

In the current paper, we analyzed only baseline values as potential predictors of cardiac events.

### 4.5. Statistical Methods

In patients receiving study drugs, the primary end-point was LVEF recovery, which was defined as a post-trastuzumab LVEF returning to no less than −5 percentage points of the baseline value.

The study was initially planned to be multi-institutional, to accrue 200 patients and to include a historical comparison cohort. We started accrual at our Institution but faced budget problems and difficulties in standardizing LVEF protocol at the other participating Institutions to minimize inter-operator variability. We, therefore, decided to proceed as a single Institution. Due to resizing, the aim of this study is exploratory, and no formal sample size calculation has been performed. The proportion of patients developing MCT is described, together with its 95% Confidence Interval (C.I.). Proportions were compared by the Chi-square test. Comparison between inter-patient and intrapatient LVEF, TN and BNP values at the baseline and at various time-points during treatment were performed by the Mann-Whitney U test and by the Wilcoxon rank test for paired samples, respectively. The Kaplan Meier method was used to describe the probability of developing MCT during treatment. Finally, uni- and multivariable logistic regression analysis was used to identify predictors of the MCT. All tests were performed by the IBM SPSS version 22 package (Chicago, IL, USA). Statistical significance was set at *p* < 0.05.

## 5. Conclusions

Randomized trials with cardiac drugs enrolling candidates to adjuvant trastuzumab-based therapy have so far provided inconclusive results. Current guidelines still recommend treating symptomatic patients only, or those that need trastuzumab withholding because of declining LVEF. Considering results, strengths and limitations, our data hint at a potential utility of targeted intervention with cardiac drugs in cases of asymptomatic, mild LVEF reductions. Furthermore, our data concur with other observations to define the need for anthracycline-free, trastuzumab-based regimens as a strategy to minimize early and late cardiac toxicity and to allow full adherence to trastuzumab treatment [19]. Being generated in the context of a mono-Institutional, non-randomized study, we believe that our data should be confirmed in a randomized clinical trial before their adoption in the clinical practice. Furthermore, the question of whether LVEF decline observed comparing pre- and post- trastuzumab values has long-term implications remains still open and relevant. To get further insights into this issue, we are planning a long-term evaluation of LVEF values in patients enrolled in this study and undergoing regular follow-up visits at our Institution.

## Figures and Tables

**Figure 1 cancers-12-00327-f001:**
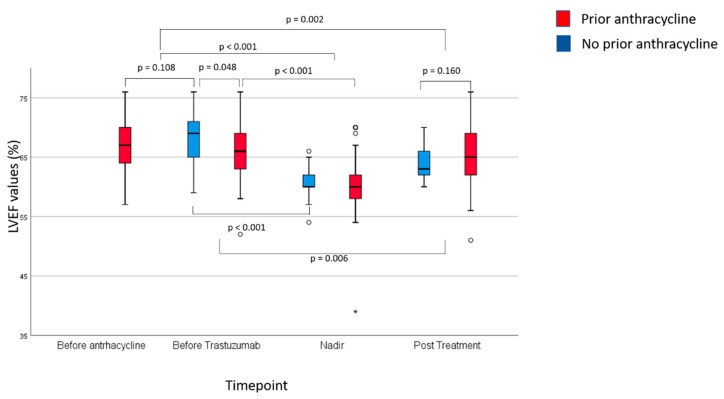
Summary of LVEF findings at baseline, LVEF nadir, and post-treatment according to prior exposure to anthracycline. The box extends from the 25th to the 75th percentile. The line is the median LVEF value. The lines extend to the largest and smallest observed values within 1.5 box lengths; “o” symbols represent outliers (values between 1.5 to 3 box lengths from the upper or lower edge of the box), asterisks represent extreme values (values of more than 3 box lengths from the upper or lower edge of the box).

**Figure 2 cancers-12-00327-f002:**
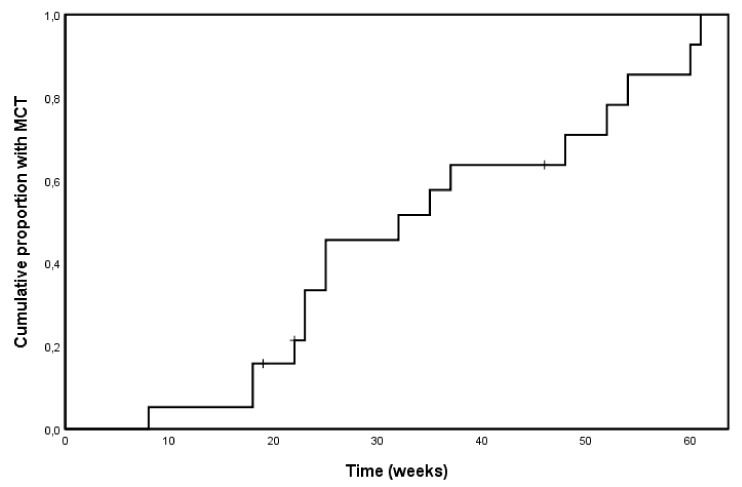
Kaplan-Meier curve of time to the development of mild cardiac toxicity (MCT).

**Figure 3 cancers-12-00327-f003:**
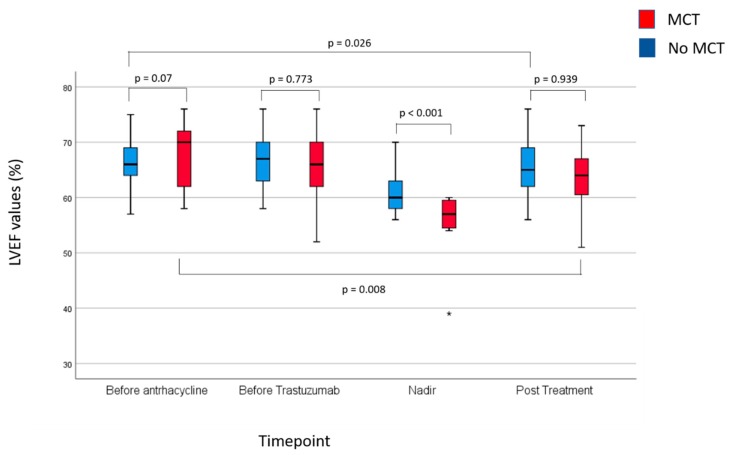
Summary of LVEF findings at baseline, LVEF nadir and post-treatment according to the occurrence of the cardiac event of interest. The box extends from the 25th to the 75th percentile. The line is the median LVEF value. The lines extend to the largest and smallest observed values within 1.5 box lengths; “o” symbols represent outliers (values between 1.5 to 3 box lengths from the upper or lower edge of the box), asterisks represent extreme values (values of more than 3 box lengths from the upper or lower edge of the box).

**Table 1 cancers-12-00327-t001:** Baseline demographics and tumor characteristics for the study population.

Characteristics	No. (%)
Median age, years (range)	53 (33–82)
Median BMI (range)	24 (17–38)
Smoking history	
Never	68 (66)
Past	6 (6)
Current	19 (18)
Not Know	9 (9)
Chronic Alcohol Exposure	5 (5)
Cardiovascular risk profile	
Hypertension	13 (13)
Type II diabetes	1 (1)
Dyslipidemia	3 (3)
Hypothyroidism	5 (5)
Hyperthyroidism	1 (1)
Stage	
I	48 (47)
II	36 (35)
III	18 (17)
Histologic subtype	
Ductal/NST	94 (91)
Lobular	6 (6)
Papillary	2 (2)
Others	1 (1)
Estrogen and/or progesterone receptor positive	77 (75)
HER2 status by IHC	
HER3+	74 (72)
HER2+/FISH positive	29 (28)
Anthracycline use in neoadjuvant or adjuvant CHT	85 (82)
Taxane use in neoadjuvant or adjuvant CHT	103 (100)
Trastuzumab administration	
Concurrent with a non-anthracycline regimen	18 (17)
Sequential, after anthracycline regimen	81 (79)
Only trastuzumab (for patient decision)	1 (1)
Did not start trastuzumab	3 (3)
Concomitant RT	59 (57)
RT on the left chest wall	30 (29)
Pre-treatment LVEF, median % (range %)	67 (57–76)
Pre-treatment BNP, median (range), pg/mL	33 (0–147)
Pre-treatment TnI, median (range), ng/mL	0.01 (0.001–0.100)

BMI: body mass index, NST: no special type, CHT: chemotherapy, RT: radiotherapy, LVEF: left ventricular ejection fraction, BNP: B-type Natriuretic peptide, TnI: troponin I.

**Table 2 cancers-12-00327-t002:** Univariate analysis of predictors of MCT.

Variable	OR	P	95% C.I.
Age (≥54)	0.586	0.587	0.253–2.176
Baseline LVEF ≥67	0.550	0.305	0.176–1.722
Left Sided RT	0.487	0.292	0.128–1.854
RT	1.250	0.691	0.416–3.757
Current or former smoker	2.143	0.513	0.218–21.047
BMI >25	0.777	0.666	0.247–2.445
Family History of CVD	3.161	0.042	1.044–9.568
Hypertension	2.778	0.131	0.738–10.642
Prior Anthracycline	4.091	0.187	0.506–33.083
Baseline TnI >0.01	1.588	0.417	0.519–4.857
Baseline BNP >33	0.886	0.827	0.299–2.620

LVEF, left ventricular ejection fraction; RT, radiation therapy; BMI, body mass index; CVD, cardiovascular disease; TnI, troponin I; BNP, b-type natriuretic peptide.

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
