# Peer review of "Treatment with Beta-Blockers and ACE-Inhibitors in Breast Cancer Patients Receiving Adjuvant Trastuzumab-Based Therapy and Developing Mild Cardiac Toxicity: A Prospective Study"

_cancers, 2020, doi:10.3390/cancers12020327_

Round 1

Reviewer 1 Report

The authors are to be congratulated for conducting this nonrandomized open interventional single group cardiac safety study in women with early stage HER2+ BC.  The authors conclude that early intervention with a beta-blocker and ACE inhibitor may be beneficial in individuals who experience early asymptomatic drops in LVEF. The authors acknowledge the limitations of this study including the lack of a control group or cohort who did not receive this intervention.

I do have some concerns with regards to the presentation of this manuscript. It is difficult to interpret the findings as presented when the results are presented before the methodology. The reader is left to try to determine the population being studied as well as the primary endpoint for this study. I would suggest the order of the manuscript be adjusted to reflect that of the abstract. 

The selection of patients needs to be further defined. How were they selected and over what period of time. Did any patients receive other HER2 targeted drugs besides trastzumab (e.g pertuzumab). What was the rationale for stopping the study drugs 3 weeks after completion of the targeted therapy? How often were patients on cardiac medication seen by the cardiologist?

The conclusion should comment further on how this data impacts what is already known and the research needed to address the current knowledge gaps. Do the authors have any further studies planned based on their findings in this study?

Minor points: 

1)The authors define mild cardiotoxicity (MCT) in the abstract but not in the body of the manuscript. Please be consistent. Is the MCT or CEI that you are addressing in this study. There is no mention of CEI in the abstract.

2) abstract: Conclusion: Treatment of MCT...

3) page 2: ...in patients with prior or previously....this sentence does not make sense

4) table 1: remove this is a table

page 4: Figure 1: please correct spelling of anthracycline 

5) please spell out CEI in the body of the manuscript the first time it appears

6) page 6: please expand on the statistically significant changes in median nadir LVEFs between the patients with and without a CEI - 2 % is statistically significant - from 60 to 58 %. Is this clinically meaningful? Also the authors have not acknowledged that these small changes can fall within the variability of the test - even when using the same machine and reader. 

7) page 7: HERA trial

8) page 7: reword this sentence. Such a small reduction......

9) page 8: ...we sought to target.....

10) page 8: I would not consider a single institution study a strength

Author Response

The authors are to be congratulated for conducting this nonrandomized open interventional single group cardiac safety study in women with early-stage HER2+ BC.  The authors conclude that early intervention with a beta-blocker and ACE inhibitor may be beneficial in individuals who experience early asymptomatic drops in LVEF. The authors acknowledge the limitations of this study including the lack of a control group or cohort who did not receive this intervention.

Comment; I do have some concerns with regards to the presentation of this manuscript. It is difficult to interpret the findings as presented when the results are presented before the methodology. The reader is left to try to determine the population being studied as well as the primary endpoint for this study. I would suggest the order of the manuscript be adjusted to reflect that of the abstract. 

R) We thank this reviewer for a positive, overall evaluation of our study. As for the order of the manuscript, it follows the template and specific requirements of the journal Cancers. We agree that results first, and methods after are not usual in clinical papers. Yet, to stick with the requirement of the journal, we had to put paragraphs in that order. In any, case, should the Editor agree, there would be no problem for us to fix the manuscript according to the referee's request.

Please, also note that part of the confusion is due to some inconsistencies that we made and that you just pointed out, as, for example, different or unclear definitions of the endpoints. As a result of incorporating your suggestions, we believe that the manuscript flows a little bit better.  

Comment; The selection of patients needs to be further defined. 1) How were they selected and over what period of time. 2) Did any patients receive other HER2 targeted drugs besides trastzumab (e.g pertuzumab). 3) What was the rationale for stopping the study drugs 3 weeks after completion of the targeted therapy? 4) How often were patients on cardiac medication seen by the cardiologist?

R)

1) We added period of patient screening to the methods section (Page 8)

2) A statement that no patient received pertuzumab added to trastuzumab has been added in the methods section (page 8)

3) Currently, there is no recommendation to treat with ACE inhibitors and beta-blockers patients with mild cardiac toxicity, as it was defined in our paper. Therefore, in order to avoid unnecessary treatment, we decided to limit the experimental treatment to completion of adjuvant trastuzumab in patients asymptomatic and with LVEF within the normal Institutional limits (≥55%). Otherwise, cardiac drugs could be continued if clinically indicated. A description of this rationale was added to the paper. Page 9.

4) A statement specifying cardiological examinations has been added to the text (Page 9).

Comment The conclusion should comment further on how this data impacts what is already known and the research needed to address the current knowledge gaps. Do the authors have any further studies planned based on their findings in this study?

R) The conclusion has been expanded as requested (page 10)

Minor points: 

1)The authors define mild cardiotoxicity (MCT) in the abstract but not in the body of the manuscript. Please be consistent. Is the MCT or CEI that you are addressing in this study. There is no mention of CEI in the abstract.

R) Indeed, the cardiac event of interest was mild cardiac toxicity (MCT). We have changed abstract, test and figure legends to keep consistency across the manuscript. 

2) abstract: Conclusion: Treatment of MCT...

R) Corrected as suggested

3) page 2: ...in patients with prior or previously....this sentence does not make sense

R) We agree! We have rephrased it in: "..in patients with prior or concomitant exposure to anthracycline-based chemotherapy."

4) table 1: remove this is a table

R) Done

page 4: Figure 1: please correct spelling of anthracycline 

R) Spelling has been fixed in the figure

5) please spell out CEI in the body of the manuscript the first time it appears

R) CEI has been substituted with mild cardiac toxicity (MCT), which is the cardiac event of interest in this study. The abstract, manuscript, figure and legends have been fixed in order to keep consistency.

6) page 6: please expand on the statistically significant changes in median nadir LVEFs between the patients with and without a CEI - 2 % is statistically significant - from 60 to 58 %. Is this clinically meaningful? Also the authors have not acknowledged that these small changes can fall within the variability of the test - even when using the same machine and reader

R) The Mann Whitney test is often misinterpreted because it does not give the significance of a difference between two medians.

It instead refers to comparing values distributions.

Thus, the medians may differ just a little bit, but the overall distribution of values may differ dramatically.

This is the case with our observation, and it can be really appreciated in figure 2. In our opinion, analytic precision can be an issue with ultrasonography, but it should not affect the significance of our finding.

We have changed the text on page 6 and, in general, avoided the use of the term "median" when referring to value comparisons. This corresponds to a better way to present comparisons by a non-parametric test like the MW (and also the Wilcoxon rank test for paired samples). We hope that this addresses the referee's comment

7) page 7: HERA trial

R) In the text we are referring to the OHERA trial, which is different from the HERA trial. OHERA was a prospective observational study of cardiac safety of adjuvant trastuzumab given after chemotherapy and published for the first time in 2019 (see reference 8). We hope that this clarifies. 

8) page 7: reword this sentence. Such a small reduction......

R) The previous sentence and the one pointed out by the referee have been rephrased to increase clarity.

9) page 8: ...we sought to target.....

R) Fixed

10) page 8: I would not consider a single institution study a strength

R) We agree, and we have erased "it was mono-Institutional" from the potential strengths of our study.

Reviewer 2 Report

This is an interesting study with results that are relevant to a significant number of patients being treated for breast cancer and support previously published studies. The overall text is somewhat cumbersome to read and the description of the primary endpoint could be better described {the way it is written (>-5%) is cumbersome and took some additional effort to interpret}; the abbreviation CEI is not defined in the text . This study is limited by being small with no comparison or control group so the addition to the literature is limited but relevant.

Author Response

Comment: This is an interesting study with results that are relevant to a significant number of patients being treated for breast cancer and supports previously published studies.

R) We are grateful to this referee for an overall positive evaluation of our manuscript.

Comment; The overall text is somewhat cumbersome to read and the description of the primary endpoint could be better described {the way it is written (>-5%) is cumbersome and took some additional effort to interpret}; the abbreviation CEI is not defined in the text.

R) We thank the referee for this comment. We have made the following changes

1) The primary end-point definition has been rephrased into: In patients receiving drug medications, the primary study end-point was LVEF recovery, which was defined as a post-trastuzumab LVEF returning to no less than -5 percentage points of the baseline value. 

We understand that the previous phrasing was cumbersome. Furthermore, it was not correct because it said >-5%, and not ≥-5%, as now more clearly states. We hope that the new phrasing is clearer. 

2) Indeed, the cardiac event of interest was mild cardiac toxicity (MCT). We have changed abstract, test and figure legends in order to keep consistency across the manuscript. 

Comment: This study is limited by being small with no comparison or control group so the addition to the literature is limited but relevant.

R) We thank the referee for acknowledging the potential impact of our data

Round 2

Reviewer 1 Report

The authors re-submitted their manuscript on the role of b-blockers and ACE-inhibitors in BC patients receiving adjuvant trastuzumab based therapy.

The authors have addressed many of the previous commnents.

Several points remain to be clarified

1) I do not agree with the statement that b-blockers and ACE inhibitors are now being used widely in this patient population upfront. Oncologists have not adopted this strategy and they are usually the gatekeeper in terms of referral to the cardiologist. Please check statement with regards to pertuzumab and trastuzumab. I think the authors meant to state this is not funded in Italy.

2) methodology: this section has improved although I would like the primary and secondary, including any exploratory objectives clearly defined. Is MCT a new definition; have previous studies been conducted with this definition.  It is not clear to me how these patients were followed – all by a cardiologist initially and then not? Were patients treated as per the ACC/AHA guidelines for heart failure.

3) results: what was the median time of follow-up for this patient population. Do we have any more detailed information on the cardiac medications taken – dose; compliance etc. What do the authors plan to do with the biomarker data – it was collected but only baseline data was reported.

4) discussion: The authors conclude that b-clockers and ACE inhibitors in patients with MCD might be an interesting approach but also acknowledge that this study DID NOT meet the primary endpoint; was small; a single institution and non-randomized. Therefore the authors should be cautious in stating that this is a reasonable approach – we clearly need more data – this study represents good pilot data for the development of a larger, multi-center, randomized control study.

4) grammar – please check throughout the document; e.g LFEV should be LVEF. Please be consistent in the use of terms ‘study’ vs ‘drug’ medication.

Author Response

We thank the referee for the opportunity to further improve our paper based on his comments. Here follows how we addressed them (bold).

1) I do not agree with the statement that b-blockers and ACE inhibitors are now being used widely in this patient population upfront. Oncologists have not adopted this strategy and they are usually the gatekeeper in terms of referral to the cardiologist. Please check statement with regards to pertuzumab and trastuzumab. I think the authors meant to state this is not funded in Italy.

R: Agreed: The statement has been erased and a recent reference reporting the ESMO guidelines has been added to the list.

Yes, Pertuzumab is not yet funded in Italy in the early setting. We erroneously omitted "not" in the phrase. We have changed it.

2) methodology: this section has improved although I would like the primary and secondary, including any exploratory objectives clearly defined. Is MCT a new definition; have previous studies been conducted with this definition.  It is not clear to me how these patients were followed – all by a cardiologist initially and then not? Were patients treated as per the ACC/AHA guidelines for heart failure.

A paragraph entitled "Study Objectives" has been added to the methods section and all the study objectives, as described in the protocol, have been listed. Longitudinal analysis of BNP and TnI fluctuations, as well as ultrasonographic parameters, are the subject of a separate publication we are currently working on.

MCT is just a reduction of LVEF that is still compatible with the continuation of trastuzumab without indication of cardiac medications.

We are not aware of other pharmacological, interventional studies using this definition. Yet, in reference 9, the potential implications of an early, mild decline (as little as 5% from baseline) are pointed out. 

Patients with MCT were identified by the study cardiologist during LVEF, placed on treatment and followed-up, during anticancer treatment, every two weeks for dose up-titration. Then they underwent regular LVEF evaluations by the cardiologist and oncology visits as requested for anticancer treatment management. If needed (i.e. symptoms, drug tolerance) they could be referred to the cardiologist at any time. This has been briefly explained in the study procedures paragraph. The Cardiac safety monitoring paragraph has been incorporated in the study procedures. 

We stated in the methods that "Patients with significant cardiotoxicity were treated according to guidelines, including trastuzumab temporary or definitive discontinuation and pharmacological interventions". Symptomatic cardiac toxicity was not a study end-point and no specific reference to a single guideline was made in the protocol. As a matter of fact, our cardiologist followed the 2013 edition of the ACC/AHA guidelines and following updates.  

3) results: what was the median time of follow-up for this patient population. Do we have any more detailed information on the cardiac medications taken – dose; compliance etc. What do the authors plan to do with the biomarker data – it was collected but only baseline data were reported.

The data cutoff for this analysis was that of the first post trastuzumab LVEF assessment, which had to be carried out between 6 and 12 months from treatment completion. This corresponded to a median of 23 months. But it is not the median follow-up. Follow-up of these patients, who were all treated and are at our Institution, is ongoing in parallel to adding new LVEF data to study long-term outcomes. Regarding the biomarker data, see previous response.

Regarding treatment, no patient had to discontinue treatment with ACEi and BB, but doses fluctuated during the study. A summary table would be very cumbersome. 

4) discussion: The authors conclude that b-clockers and ACE inhibitors in patients with MCD might be an interesting approach but also acknowledge that this study DID NOT meet the primary endpoint; was small; a single institution and non-randomized. Therefore the authors should be cautious in stating that this is a reasonable approach – we clearly need more data – this study represents good pilot data for the development of a larger, multi-center, randomized control study.

The final statement has been changed in: "Considering results, strengths and limitations, our data hint at a potential utility of targeted intervention with cardiac drugs in cases of asymptomatic, mild LVEF reductions"

4) grammar – please check throughout the document; e.g LFEV should be LVEF. Please be consistent in the use of terms ‘study’ vs ‘drug’ medication.

Thank you for this. Indeed, it was a little bit messy. We have fixed this issue.